# Bioaccumulation of Heavy Metals in a Soil–Plant System from an Open Dumpsite and the Associated Health Risks through Multiple Routes

Muhammad Sabir [1], Edita Baltrėnaitė-Gedienė [2], Allah Ditta [3,4,*], Hussain Ullah [5], Aatika Kanwal [1], Sajid Ullah [6] and Turki Kh. Faraj [7]

[1] Department of Environmental Sciences, University of Haripur, Haripur 22620, Pakistan
[2] Department of Environmental Protection, Vilnius Gediminas Technical University, 10223 Vilnius, Lithuania
[3] School of Biological Sciences, The University of Western Australia, Perth, WA 6009, Australia
[4] Department of Environmental Sciences, Faculty of Sciences, Shaheed Benazir Bhutto University, Sheringal 18000, Pakistan
[5] Department of Chemistry, University of Okara, Okara 56300, Pakistan
[6] School of Resource and Environmental Engineering, East China University of Science and Technology, Shanghai 200237, China
[7] Department of Soil Science, College of Food and Agricultural Sciences, King Saud University, Riyadh 11451, Saudi Arabia
* Correspondence: allah.ditta@uwa.edu.au or allah.ditta@sbbu.edu.pk

**Abstract:** Screening various plant species to act as hyperaccumulators and associated health risks could serve as a sustainable solution for the bioremediation heavy metals (HMs). For the first time, the present study explored the phytoremediation potential of native plants, soil enrichment, and human health risks associated with the contamination of HMs in soil and plant samples collected from a municipal solid-waste open dump site. Soil and plant samples ($n = 18 + 18$) from the dumpsite and ($n = 18$) from the control site were analyzed for selected HMs, i.e., Chromium (Cr), Lead (Pb), Nickel (Ni), Iron (Fe), and Zinc (Zn). The phytoremediation potential of plants was assessed using the bioaccumulation factor (BAF), bioaccumulation coefficient (BAC), and translocation factor (TF), while soil pollution levels were evaluated using the contamination factor (CF), geoaccumulation index (Igeo), enrichment factor (EF), potential ecological risk index (PERI), and human health risk indices. The results revealed that based on TF and BAC values, *Alhagi maurorum* Medic., *Astragalus creticus* Lam., *Cichorium intybus* L., *Berberis lycium* Royle, and *Datura stramonium* L. were hyperaccumulators for Cr while *Parthenium hysterophorus* L. was a promising species for both Ni and Cr. Similarly, CF values for Fe, Ni, Pb, and Cr were >6, thereby showing very high contamination, while Igeo values for Fe, Ni, Pb, and Cr were (class 6, >5), showing that the soil was extremely polluted. Furthermore, EF values for Fe, Ni, Pb, Cr, and Zn were $2 < EF \leq 5$, depicting moderate enrichment, while PERI values were in the range of 91.31–195.84, employing moderate ecological risks ($95 < PERI < 190$) from the dumpsite's soil. Moreover, for non-carcinogenic exposure, none of the analyzed metals exceeded the threshold limit HRI values > 1 in both adults and children. Likewise, in the case of carcinogenic effects, the CRI values were lower than the tolerable limits ($1 \times 10^{-6}$–$1 \times 10^{-4}$) in both adults and children. Moreover, almost all studied plants could be utilized for the phytoextraction of mentioned HMs. In future, the present study can help in the implementation of public policies to ensure sustainability and developmental activities in contaminated sites. Based on these results, it is concluded that there is a dire need of monitoring solid waste dumpsites due to various types of potential risks associated with the contamination of HMs. Moreover, to minimize the potential health problems arising from the dumpsite, it is substantive that special attention should be paid to work on sustainable and eco-friendly remedial measures.

**Keywords:** heavy metals; open dumpsite; bioaccumulation of metals; human health risk; geoaccumulation; phytoremediation

## 1. Introduction

Environmental pollution due to anthropogenic and geogenic origins has significant negative impacts on biota [1]. Rapid industrialization, urbanization, population growth, and economic development increased the production of the amount of solid waste per capita [2]. The increased solid waste generation has led to open dumpsites that are known to be the most common and oldest methods for solid waste disposal in developing countries such as Pakistan [3]. Increased waste generation, growing population, accelerated urbanization, limited capital, socioeconomic inequalities, communal expectations, and inadequate legislation have all contributed to the system's complexities and poor solid waste management. With a population of 230.7 million people, Pakistan is the second largest country in South Asia and the sixth in the world; however, unfortunately, Pakistan's solid waste management situation is alarming [4]. In particular, in developing countries with financial constraints, solid waste disposal is far from the standard recommendation, thus causing serious threats to the environment. Solid waste from fossil fuel burning, municipal waste, mining activities, fertilizers, and pesticides is reported as the major source of heavy metals (HMs) [5].

The accumulation of HMs in soil due to open dumping sites is one of the most emergent environmental problems in developing countries [6]. A higher concentration of HMs in soil induces toxicity and retard plant growth. Due to their persistent nature, high toxicity, bioavailability in the open environment, and the potential for greater bioaccumulation and biomagnification have posed severe health risks to humans, animals, and plants [7]. For instance, lead (Pb) can adversely affect the human central nervous system (CNS); can cause abdominal pain, irritability, sleeplessness, and headache; and can cause behavioral abnormalities and learning issues in children under the age of five due to higher susceptibility [8]. A higher concentration of zinc (Zn) may cause infertility, CNS disorder, and kidney disease. Chromium (Cr) is highly carcinogenic and may result in damage to the respiratory system. Excessive iron (Fe) concentration causes molybdenosis in living organisms [9]. Hence, it is important to keep checking and balancing HM concentrations in different media of the environment in order to avoid toxic effects on biota.

Several technologies such as adsorption, chlorination, chemical extraction, ion exchange, electrokinetic, bioleaching, thermal treatment, phytoremediation, and bioremediation have been discovered by scientists to remediate soils contaminated with HMs [10]. However, phytoremediation is the most economical, practical, and eco-friendly approach for HMs remediation among all the available methods [11]. Phytostabilization and phytoextraction are the two most common techniques of phytoremediation. Phytoextraction involves the use of native plants to accumulate HMs in shoot and root portions, which can be removed through harvesting. Plants that can be used for phytoextraction should be fast growing with high biomass production potentials, highly branched root systems, be widely distributed, and be easily cultivated and harvested [12]. Furthermore, an ideal plant for phytoextraction must have a translocation factor (TF) and bioaccumulation concentration (BAC) values > 1. As per Brook and Baker [13], hyperaccumulators plants can accumulate 100 mg kg$^{-1}$ of Cd, 1000 mg kg$^{-1}$ of Cu (copper), As (Arsenic), Pb, Ni (Nickel), Co (Cobalt), Se (selenium), and Cr and 10,000 mg kg$^{-1}$ of Mn (manganese) and Zn. Plants with high metal tolerance, low metal transported rate, and increased microbiological diversity are considered key candidates for phytostabilization.

Native plants could be effectively utilized for phytoremediation because they can perform better in terms of survival, growth, and reproduction [14]. Various studies have been conducted in Pakistan to assess hyperaccumulator plants grown around mining sites, ophitic belt zone, roadside, and plants irrigated with wastewater [5–7,10]. However, few or no studies have been conducted to screen hyperaccumulator plants growing around the municipal solid-waste open dump site. Furthermore, there is no previous data available regarding potential ecological and human health risk assessment data from open dumpsites in Pakistan and particularly in the study area. These multiple route exposure (inhalation, dermal absorption, and direct ingestion) data are necessary to thoroughly assess the dump-

site's deleterious impacts on human health. Moreover, the contamination of the soil-plant continuum with heavy metals such as Cr, Ni, Fe, Pb, and Zn has also been in focus in recent studies [7–12]. Taking these problems into consideration and to find out a sustainable solution in the form of screening various plant species to act as hyperaccumulators for different heavy metals, the current study was conducted to screen hyperaccumulator plants growing around the solid waste dumpsite of Peshawar and to assess HM concentrations in the dumpsite's soil and plants and their associated risks to human health.

## 2. Materials and Methods

### 2.1. Research Area

Peshawar is the capital city of Khyber Pakhtunkhwa, a province of Pakistan. The city is located near the eastern end of Khyber Pass, close to the border of Afghanistan. Geographically, the study area is situated between $33°57'40''$ and $33°58'0''$ north latitude and $71°34'30''$ and $71°34'45''$ east longitude (Figure 1).

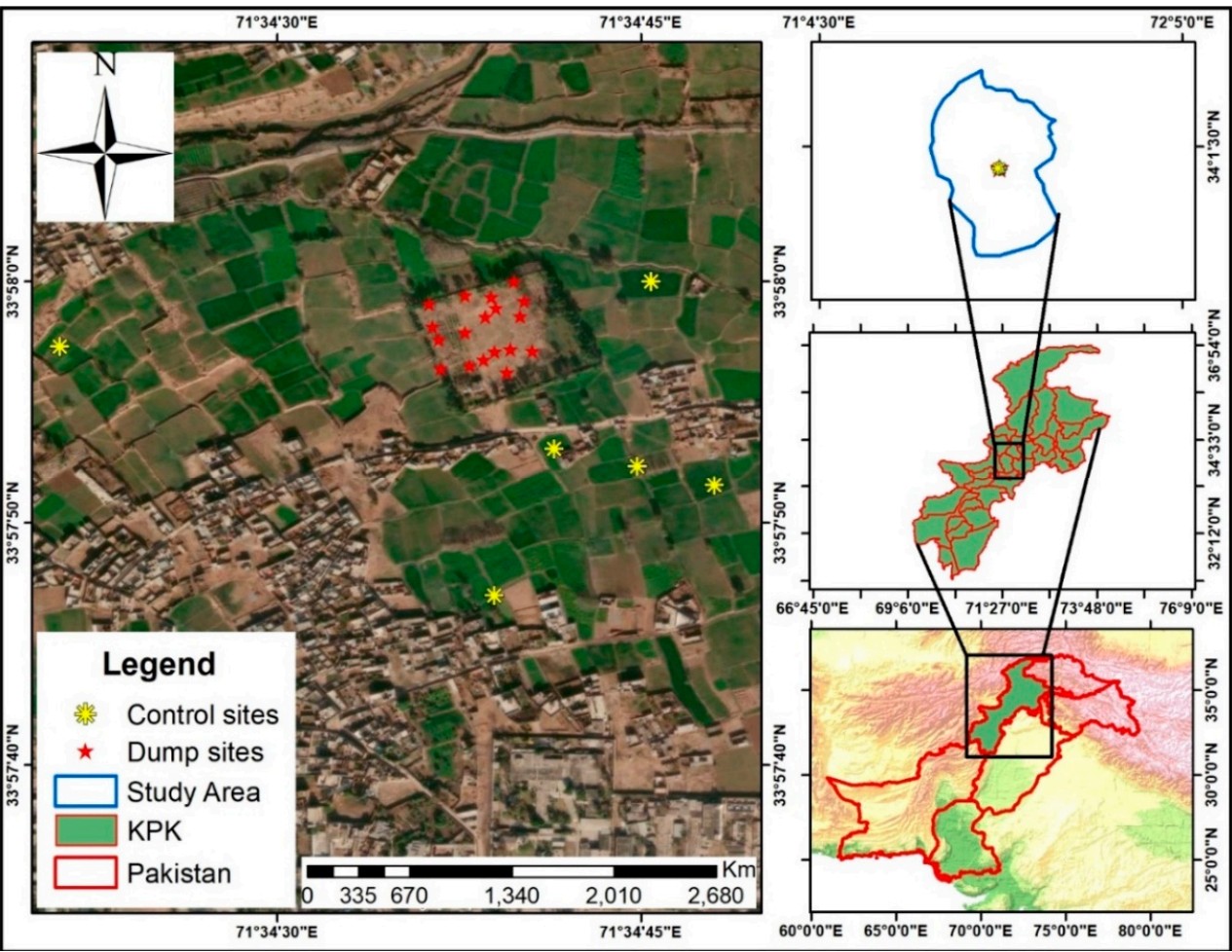

**Figure 1.** Map of the study area.

The total area of the city is 1275 km$^2$, with an approximate population of four million, and the city is divided into four towns and phases [15]. The study area is populous, so municipal wastes, agriculture waste, household garbage, food waste, and other industrial waste are being disposed of without proper physical and chemical treatments in open dump sites. The selected dumpsite is the major among all and occupied 0.3 square kilometers in the heart of the city and is in use since 1997, thus obeying the criteria used for risk assessment studies.

### 2.2. Sample Collection and Pre-Treatment

Soil and plant samples in triplicate ($n$ = 18 + 18 for each) from the dumpsite and ($n$ = 18 for each) from the control site were collected based on the species' dominance. Random and composite soil sampling techniques were used, and the soil samples were collected from a depth of 0–25 cm with stainless steel augur, as directed by Ahmad et al. [16]. Then, the collected soil samples were transported to the laboratory in polyethylene bags and stored in the refrigerator until analyses.

### 2.3. Samples Analytical Procedure

The soil samples were ground with the help of an agate mortar, sieved, and dried in an oven at 80 °C until a constant weight was obtained [17]. Physiochemical parameters such as pH, EC, texture, organic matter, and heavy metals (HMs) were determined by following the standard procedures of Ullah et al. [18]. Soil pH and EC were determined using a soil water mixture of 1:10 $w/v$ [19]. Soil TDS, porosity, and bulk density were found using the protocol followed by Sabir et al. [20]. For HMs analysis, a tri acidic solution was prepared in which $HClO_4$, $HNO_3$, and $H_2SO_4$ were mixed in a 1:5:10 ratio, respectively. Then, 0.5 g of each soil sample was added to 15 mL Aqua Regia and left overnight. The next day, the soil samples were carried to a digestion block for complete digestion. The mixture was heated up to 80 °C for 1 h, and then the temperature was raised to 160–180 °C until a transparent solution was obtained. Next, the transparent solution was cooled, filtered through a 45 μ Whatman paper, and diluted up to 50 mL through distilled water for HM analysis [21–23]. Similarly, plant samples were washed with tap water and distilled water. Each plant was divided into root and shoot, air dried, followed by oven drying, and powered with pestle and mortar. The same analytical procedure was repeated for plants for HMs analysis as was followed for soil samples. To ascertain the reliability of the analytical data, blank samples were prepared and a quality control sample was run after every seven measurements. The ICP-OES was set at a UV exposure time of 20 s; UV neb gas flow of 0.5 L min$^{-1}$; UV RF power of 1150 W; VIS exposure time at 5 s; and the cool gas flow rate was 0.5 L min$^{-1}$. All metals were measured using the axial mode at wavelengths (nm) as follows: Cr = 285.546, Pb = 230.543, Zn = 254.567, and Ni = 243.587.

### 2.4. Phytoremediation Potential of Plants

The bioaccumulation potential of the studied plants was calculated using the following formulas [24].

$$\text{Bioaccumulation factor} = \frac{\text{Heavy metals in roots} \left( \text{mg kg}^{-1} \text{ DW} \right)}{\text{Heavy metals in soil} \left( \text{mg kg}^{-1} \text{ DW} \right)}$$

$$\text{Bioaccumulation coefficient} = \frac{\text{Heavy metals in the shoot} \left( \text{mg kg}^{-1} \text{ DW} \right)}{\text{Heavy metals in soil} \left( \text{mg kg}^{-1} \text{ DW} \right)}$$

$$\text{Translocation factor} = \frac{\text{Heavy metals in the shoot} \left( \text{mg kg}^{-1} \text{ DW} \right)}{\text{Heavy metals in root} \left( \text{mg kg}^{-1} \text{ DW} \right)}$$

### 2.5. Assessment of Heavy Metals Pollution

### 2.5.1. Contamination Factor

The contamination factor is the ratio of targeted metal in the soil to its background value, as provided by Hakanson [25]. He classified the contamination values as CF < 1 (low contamination), 1 < CF < 3 (moderate contamination), 3 < CF < 6 (considerable contamination), and CF > 6 (very high contamination). The background values taken for Cr, Fe, Pb, Ni, and Zn were 90, 900, 35, 750, and 175 mg kg$^{-1}$, respectively [25].

$$\text{Contamination factor} = \frac{\text{Concentration of heavy metal in the sample} \left(\text{mg kg}^{-1} \text{ DW}\right)}{\text{Background value of the heavy metal} \left(\text{mg kg}^{-1} \text{ DW}\right)}$$

### 2.5.2. Geoaccumulation Index

The geoaccumulation index (Igeo) could be instrumental for formulating effective environmental planning. In the current study, it was developed to find HM's contamination status of dumpsite soil and to identify natural and anthropogenic sources. Igeo was assessed by the below-mentioned equation, where $C_n$ (mg kg$^{-1}$ DW) represents the measured concentration of metals, i.e., n and $B_n$ (mg kg$^{-1}$ DW) are the geochemical background value of the corresponding metal and 1.5 is the compensation factor in the background concentration of HMs. In the current study, shale values were used as background values. Hakanson [25] classified the Igeo index as "unpolluted (Igeo $\leq$ 0), unpolluted to moderately polluted (0 < Igeo $\leq$ 1), moderately polluted (1 < Igeo $\leq$ 2), moderately to heavily polluted (2 < Igeo $\leq$ 3), heavily polluted (3 < Igeo $\leq$ 4), heavily to extremely polluted (4 < I geo $\leq$ 5)".

$$I_{geo} = \log_2 \left[ \frac{C_n}{1.5 \times B_n} \right]$$

### 2.5.3. Enrichment Factor

The enrichment factor (EF) holds equal importance through which we can assess the level of HM pollution from anthropogenic sources and calculate it using the following equation.

$$\text{Enrichment factor} = \frac{\text{sample (Metal/Fe)}}{\text{Background (Metal/Fe)}}$$

For geochemical normalization, Fe concentration (mg kg$^{-1}$ DW) was used as a reference metal because it is mostly found in combination with very fine surface solids. Secondly, Fe geochemistry is similar to most trace elements, and naturally, it occurs uniformly in the environment. EF values were interpreted as EF < 1 (no enrichment), 1 < EF < 3 (minor enrichment), 3 < EF < 5 (moderate enrichment), 5 < EF < 25 (severe enrichment), 25 < EF < 50 (very severe enrichment), and EF > 50 (extremely severe enrichment) [26].

### 2.5.4. Degree of Contamination

Hakanson [25] presented an investigation tool for simplifying pollution control using degrees of contamination (DC). He proposed a classification for DC as DC < 6 (low degree of contamination), 6 < DC < 12 (moderate degree of contamination), and 12 < DC < 24 (considerable degree of contamination) and used them as indications of alarming anthropogenic contamination. It was designated as the summation of CF of each element concerned.

$$DC = \sum n \, CF$$

### 2.5.5. Potential Ecological Risk

The proposed ecological risk model assesses the class of pollution in soil primarily based upon HMs' toxicity and environmental response. The techniques encompass a range

of disciplines for biotoxicity for assessing ecological threats triggered by toxic metals. The main function of the model was to prioritize metals as per their toxicity [27].

$$E_{ri} = T_{ri} \times CF$$

$$PERI = \sum_{f=1}^{n} E_{ri}$$

In the above equation, PERI can be determined as "the summation of all risk value posed by HMs in soil", while $E_{ri}$ shows the monomial ecological risk value, CF is taken as the contamination factor, and $T_{ri}$ represents the toxic or lethal response value. It was established to calculate the potential threat due to HMs by describing the threshold limit and to find the extent to which the environment is sensitive to the corresponding metals. The $T_{ri}$ values for Cr, Ni, Zn, and Pb were 2, 5, 5, and 5, respectively. Hakanson [25] proposed the following terms for $E_{ri}$ interpretation: Eri < 40 (low ecological risk), 40 < $E_{ri}$ < 80 (moderate ecological risk), 80 < $E_{ri}$ < 160 (considerable ecological risk), 160 < $E_{ri}$ < 320 (high ecological risk), and $E_{ri}$ > 320 (very serious ecological risk). In the same manner, PERI was described as PERI < 95 (low ecological risk), 95 < PERI < 190 (moderate ecological risk), 190 < PERI < 380 (considerable ecological risk), and PERI > 380 (very high ecological risk).

2.5.6. Average Daily Dose

For the estimation of metal exposure, the average daily dose (ADD) through different exposure routes and estimated lifetime ADD (LADD) were computed for each metal path interaction using the following equation [28]:

$$ADD_{ing} = C\left(\frac{IngR \times EF \times ED \times CF}{AT \times BW}\right)$$

$$ADD_{inh} = C\left(\frac{InhR \times EF \times ED}{AT \times PEF \times BW}\right)$$

$$ADD_{derm} = C\left(\frac{IngR \times SA \times EV \times EF \times ED \times AF \times ABS \times CF}{AT \times BW}\right)$$

$$LADD = C\left(\frac{CR \times EF \times ED \times AF \times ABS \times CF}{PEF \times AT \times BW}\right)$$

where C represents the HM concentration in mg/kg in the dumpsite's soil. For adults, IngR (ingestion rate) = 100 mg day$^{-1}$, InhR (inhalation rate) = 16 m$^3$ day$^{-1}$, (body weight) = 65.7 kg, ED (exposure duration) = 24 years, and SA (skin surface area parameter) = 5000 cm$^2$, while the said parameters for children are 200 mg day$^{-1}$, 10.3 m$^3$ day$^{-1}$, 15 kg, 10 years, and 1800 cm$^2$, respectively. Likewise, AF (adherence factor of soil to skin) = 1 mg cm$^{-2}$, EF (exposure frequency) = 350 day year$^{-1}$, EV (event frequency) = 1 event day$^{-1}$, ABS (dermal absorption factor) = 0.001 (unit less), PEF (particular emission factor) = 1.32 $\times$ 10$^9$ m$^3$ kg$^{-1}$, and FC (factor for conservation) = 1 $\times$ 10$^{-6}$ mg kg$^{-1}$. Similarly, AT (average time) for non-carcinogens was ED $\times$ 365 days year$^{-1}$, while for carcinogens, it was AT = 70 $\times$ 365 = 25,550 days. In the current study, all reference doses (RfDs), slope factors (SFs), and other calculated parameters used are based on previous studies conducted in the same context [29–33].

### 2.5.7. Non-Carcinogenic Risks

Each receptor group was assessed for each HM exposure from soil to determine non-carcinogenic risk levels. For this purpose, the estimated ADD for each HM was taken as the numerator while the corresponding RfD was taken as the denominator, where RfD represents the maximum permissible hazards through everyday exposure for each metal. For instance, in the current study, for children and adults throughout their lifetime, HRI < 1 indicates ADD < RFD, which suggests no antagonistic effects. However, if HRI > 1, it means a higher ADD value than the RFD and, consequently, might have an antagonistic influence on human health. For non-cancer-causing agents, a threshold limit has been established underneath, and there is no harmful response. In the current study, the RfD values used are reported by different researchers [34–36].

The summation of HRI for three major pathways of HMs, namely, oral ingestion, nasal inhalation, and skin contact, provides a hazard index (HI) as expressed where HI < 1 indicates a safe range, while an HI greater than 1 designates the potential of non-carcinogenic risks [30].

$$\text{HRI} = \frac{\text{ADD}}{\text{RfD}}$$

$$\text{HI} = \sum \text{HRI}$$

### 2.5.8. Carcinogenic Risk Index from Soil

Both groups, i.e., children and adults, were considered for the exposure assessment of carcinogens (Cr, Ni, and Pb) by ingestion pathways. The incremental lifetime cancer risk (CRI) was calculated by utilizing the corresponding ADDs. The computed value for the cancer slop factor (SF) employs the possibility of cancer development for every exposure event (mg kg$^{-1}$ day$^{-1}$). The SF values for Cr, Pb, and Ni used in the current study were as $4.20 \times 10^1$, $8.5 \times 10^{-3}$, and $8.40 \times 10^{-1}$ mg kg$^{-1}$ day$^{-1}$, respectively. If the value of CRI is lower than $1 \times 10^{-6}$, then the risk factor is likely more imminent. Moreover, if the CRI value lies between $1 \times 10^{-6}$ and $1 \times 10^{-4}$, it demonstrates an acceptable or tolerable risk for human health [37].

$$\text{CRI} = \frac{\text{LADD}}{\text{SF}}$$

### 2.6. Statistical Analysis

The data were analyzed by using the statistical package for social sciences SPSS v. 20.0 (International Business Machines Corporation, IBM, Armonk, NY, USA). Pearson's correlation test assessed the statistical significance at $p < 0.05$ by using PAST v. 4.0.1. Principle component analysis (PCA) was used to find out associations and sources of HMs in plants and soil. ArcGIS 10.8.1 was used for mapping purposes.

## 3. Results and Discussion

### 3.1. Physiochemical Characteristics of Soil

Soil contamination with heavy metals (HMs) and metalloids was ubiquitous and considerably increased over the past few decades with rapid industrialization thereby conferring serious risks to the environment and human health [38]. HM concentrations in soil mainly depends upon soil pH, moisture, organic matter, and texture [39]. Table 1 shows the detailed information about the physicochemical information of soil collected from the solid waste dumpsite and control site. In the dump site's soil, pH, EC, TDS, OM, porosity, bulk density, and texture were $6.9 \pm 2.6$, $700 \pm 340$ µS cm$^{-1}$, $170 \pm 65$ mg L$^{-1}$, $18.42 \pm 5.54$ mg kg$^{-1}$, $80 \pm 19\%$, $3.54 \pm 1.43$ g cm$^{-3}$, and sandy loam, while in the control site, the values were as $7.1 \pm 0.87$, $180 \pm 98$ µS cm$^{-1}$, $44 \pm 11$ mg L$^{-1}$, $0.65 \pm 0.14$ mg kg$^{-1}$, $21 \pm 7\%$, $1.02 \pm 0.33$ g cm$^{-3}$, and silty clay loam, respectively.

**Table 1.** Physicochemical characteristics of dumpsite and control site soil of the study area.

| Parameters | Observations | Dumpsite (Mean ± SD) | Observations | Control Site (Mean ± SD) |
|---|---|---|---|---|
| pH | 18 | 6.9 ± 1.6 | 6 | 7.1 ± 0.87 |
| EC ($\mu$S cm$^{-1}$) | 18 | 700 ± 40 | 6 | 180 ± 28 |
| TDS (mg L$^{-1}$) | 18 | 170 ± 15 | 6 | 44 ± 5 |
| OM (mg kg$^{-1}$) | 18 | 18.42 ± 1.54 | 6 | 0.65 ± 0.04 |
| Porosity (%) | 18 | 80 ± 9 | 6 | 21 ± 2 |
| Bulk density (g cm$^{-3}$) | 18 | 3.54 ± 0.43 | 6 | 1.02 ± 0.03 |
| Texture | 18 | Sandy loam | 6 | Silty clay loam |

Where SD = standard deviation (*n* = 3).

Likewise, HMs concentrations in the dumpsite were in the range of 784–1234, 543–894, 1322–1643, 879–1368, and 445–879 mg/kg with mean values of 980 ± 230, 742 ± 180, 1465 ± 163, 1168 ± 256, 660 ± 217 mg kg$^{-1}$ for Cr, Ni, Fe, Pb, and Zn, respectively (Table 2). While in the control site, Cr, Ni, Fe, Pb, and Zn ranged at 21–56, 23–78, 104–327, 19–76, and 19–287 mg kg$^{-1}$, respectively. A significant difference ($p < 0.05$) was observed between the HM concentration in plants and soil samples collected from the dumpsite to that of the control site. In the dumpsite, all analyzed metals were above the permissible ranges given by US-EPA [40]. The elevated metal concentration could be attributed to the industrial waste disposal in the studied dumpsite. For instance, during Pb-acid battery recycling, different processes such as crushing, fusion, refining, reduction, etc., release different species of Pb in the form of anglesite (PbSO$_4$), cerussite (PbCO$_3$), metallic lead (Pb), and Pb oxide (PbO), ultimately landing in the dumpsite [41]. Similarly, higher Zn contents were observed in the dumpsite's soil, which might be because of the disposal of bottle caps, blades, and different pharmaceutical leftovers.

Ni pollution is widely distributed around the world due to its abundance in soil. Elevated Ni and Fe concentrations could be attributed to vehicular exhaust, agriculture fertilizer, incinerated hospitals, municipal waste, and other industrial waste disposals at the dump site. A likely explanation for the higher bioavailability of the HMs is the soil chemistry of the dump soil. Sorption–desorption reactions strongly influence HMs' mobility and bioavailability in soil [42]. Soil physiochemical properties have a stronger influence on HMs bioavailability. For instance, soil OM can significantly influence metal behavior by binding with toxic metals, thereby alleviating metal toxicity in soil [43]. Earlier, Chandra and Kumar [44] mentioned that soil with higher OM contents showed higher Pb contents than Cd in the control environment, which demonstrates that Pb has higher affinity and stability towards OM than Cd. The difference in relative binding affinities among the metals is mainly because of soil chemistry. In the same manner, soil pH was slightly acidic to basic in the dumpsite's soil. Soil pH holds a significant influence on the bioavailability of HMs in different media and their subsequent toxic effects on biota. In low pH soil, the mobility and bioavailability of HMs are greater compared to soil with high pH [45]. Similarly, similarly to the mobility and bioavailability of metal, pH plays a significant role in metal speciation in soil. For instance, Cr-OM complexes can affect the bioavailability of metals. Similarly to other factors, soil texture is one of the deciding factors that induce metals bioavailability in soil. It is reported that crops grown on sandy soil are more metal deficient than those grown on soil with a loamy texture, which is likely to have low metal retention capacity. A very close association was found between soil texture and HM concentrations [46].

**Table 2.** Descriptive statistics of heavy metals in dumpsite and control site soils and plants.

| Plants Species | | Cr | | | Ni | | | Fe | | | Pb | | | Zn | | |
|---|---|---|---|---|---|---|---|---|---|---|---|---|---|---|---|---|
| | | Soil | Root | Shoot | Soil | Root | Shoot | Soil | Root | Shoot | Soil | Root | Shoot | Soil | Root | Shoot |
| *A. creticus* Lam. | Control | 21 ± 5 | 13 ± 2 | 9 ± 1 | 24 ± 5 | 34 ± 7 | 22 ± 4 | 104 ± 25 | 78 ± 11 | 66 ± 10 | 165 ± 20 | 76 ± 15 | 43 ± 9 | 19 ± 3 | 29 ± 7 | 7 ± 0.5 |
| | Dumpsite | 315 ± 154 | 118 ± 65 | 46 ± 27 | 190 ± 71 | 28 ± 24 | 15 ± 9 | 1465 ± 163 | 525 ± 270 | 251 ± 127 | 499 ± 206 | 236 ± 100 | 43 ± 19 | 62 ± 7 | 46 ± 18 | 24 ± 7.16 |
| *A. maurorum* Medic. | Control | 56 ± 11 | 39 ± 7 | 14 ± 3 | 45 ± 5 | 6 ± 0.3 | 3 ± 0.1 | 132 ± 14 | 45 ± 4 | 13 ± 1 | 432 ± 32 | 56 ± 12 | 12 ± 2.5 | 87 ± 9 | 23 ± 4 | 4 ± 0.1 |
| | Dumpsite | 268 ± 153 | 106 ± 71 | 26 ± 7 | 365 ± 90 | 121 ± 78 | 22.3 ± 13 | 580.3 ± 98 | 231 ± 38 | 88 ± 24 | 550 ± 540 | 445 ± 151 | 206 ± 97 | 386 ± 163 | 160 ± 123 | 56 ± 50 |
| *P. hysterophorus* L. | Control | 276 ± 35 | 156 ± 14 | 43 ± 8 | 98 ± 11 | 24 ± 5 | 11 ± 2 | 112 ± 22 | 44 ± 7 | 23 ± 3 | 65 ± 9 | 23 ± 4 | 12 ± 2 | 143 ± 22 | 56 ± 13 | 21 ± 2 |
| | Dumpsite | 513 ± 157 | 301.6 ± 85 | 59 ± 22 | 226 ± 97 | 133 ± 90 | 85.66 ± 66 | 575 ± 204 | 345 ± 111 | 133 ± 87 | 360 ± 125 | 191 ± 85 | 89 ± 42 | 587 ± 206 | 261 ± 111 | 92 ± 48 |
| *B. lycium* Royle | Control | 349 ± 37 | 132 ± 13 | 47 ± 7 | 132 ± 12 | 76 ± 20 | 34 ± 23 | 225 ± 40 | 84 ± 12 | 45 ± 8 | 376 ± 22 | 235 ± 14 | 83 ± 7 | 132 ± 19 | 56 ± 6 | 13 ± 3 |
| | Dumpsite | 1184 ± 295 | 419 ± 128 | 128 ± 33 | 594 ± 218 | 273 ± 164 | 99 ± 43 | 780 ± 209 | 293 ± 156 | 108 ± 91 | 1140 ± 308 | 562 ± 106 | 117 ± 36 | 735 ± 260 | 448 ± 111 | 137 ± 15 |
| *D. stramonium* L. | Control | 325 ± 33 | 54 ± 8 | 13 ± 2 | 365 ± 50 | 69 ± 13 | 32 ± 7 | 327 ± 40 | 34 ± 12 | 10 ± 2 | 435 ± 32 | 78 ± 10 | 23 ± 5 | 287 ± 14 | 43 ± 5 | 3 ± 0.5 |
| | Dumpsite | 623 ± 146 | 308 ± 156 | 55 ± 38 | 742 ± 180 | 305 ± 168 | 81 ± 29 | 707 ± 177 | 195 ± 131 | 69 ± 14 | 1168 ± 256 | 560 ± 178 | 60 ± 33 | 555 ± 220 | 304 ± 162 | 61 ± 15 |
| *C. intybus* L | Control | 347 ± 140 | 221 ± 95 | 54 ± 20 | 365 ± 135 | 23 ± 8 | 9 ± 3 | 287 ± 150 | 154 ± 65 | 90 ± 25 | 376 ± 153 | 210 ± 98 | 87 ± 32 | 236 ± 85 | 76 ± 20 | 23 ± 8 |
| | Dumpsite | 980 ± 230 | 388 ± 158 | 88 ± 30 | 594 ± 173 | 288 ± 126 | 153.6 ± 52 | 754.6 ± 390 | 459 ± 227 | 107 ± 44 | 783 ± 214 | 411 ± 217 | 141 ± 27 | 660 ± 217 | 296 ± 150 | 185 ± 124 |

The values following ± represent standard deviation (*n* = 3).

### 3.2. Heavy Metals in Plants

The soil-plant transfer of metals and nutrients is natural and a part of the nutrient cycle [47]. Metals are taken up in different concentrations by plants, most often through soil solutions, and higher metal accumulation indicates higher metal contents in soil. Regarding the potential contamination and toxicity to biota, some metals such as Zn, Cu, Mn, Mo, and Ni are essential at low concentrations. However, higher accumulation than the threshold limits can cause serious damage [48]. In the current study, five HMs, i.e., Cr, Pb, Ni, Zn, and Fe, were analyzed in the selected plant roots and shoots (Table 2). Cr concentrations in the analyzed plant roots and shoots ranged from 56 to 567 and 13 to 165 mg kg$^{-1}$, and the highest Cr concentration was observed in *B. lycium* Royle, while the lowest concentration was in *A. creticus* Lam. Cr concentrations in the dumpsite's plants was significantly ($p < 0.05$) higher than in the control site's plants. Cr is potentially toxic at a higher concentration for the normal growth and development of plants. The concentration of heavy metals in plant samples was higher in comparison to the other studies [18]. Toxicity occurs due to the mutagenic and inhibitory impact of heavy metals on enzymatic activities, which ultimately results in reduced root growth and low yields [9,13].

Elevated Cr contents can remarkably decrease water potential in leaf air spaces, adversely affecting the transportation rate of different nutrients in plants and reducing Fe accumulation, total protein, chlorophyll contents, and CAT activity in plants [49].

Ni concentrations in plant roots and shoots ranged from 20 to 453 and 13 to 214 mg kg$^{-1}$. *D. stramonium* L. accumulated the highest concentration, while *A. creticus* Lam accumulated the lowest concentration of Ni (Table 2). Ni is an essential trace element for plants when available within threshold limits; however, its excess can render toxicity symptoms such as necrosis and chlorosis in different plant species. It is known to cause toxicity and affects protease and ribonucleic enzyme activities, which can lead to retarded seed germination and crop growth [50]. It has also been reported that Ni concentrations equally influence the mobilization and digestion of carbohydrates and proteins in germinating seeds, consequently decreasing root length, plant height, pigment contents, and fresh and dry weight and increasing malondialdehyde contents and electrolyte leakage. The elevated concentration of Ni affects photosynthetic pigments and can decrease water potential and anti-oxidative enzymes, $H_2O_2$ contents, lipid peroxidation, and proline levels [51].

Fe showed varied concentrations in plant roots and shoots and ranged from 98 to 4689 and 45 to 356 mg kg$^{-1}$, respectively (Table 2). The highest and lowest Fe concentration was observed in *C. intybus* L. and *D. stramonium* L., respectively. Fe plays an important role in plant photosynthetic activities [45]. Fe phytotoxicity normally exists in the form of bronzing and stippling of plant leaves. It is required for key biological functions such as photosynthesis, nitrogen fixation, sulfur assimilation, hormones, DNA synthesis, and mitochondrial respiration. However, Fe is a very abundant element in Earth's crust but is very poorly available to the plants under oxidative and alkaline conditions. Fe concentration >500 mg kg$^{-1}$ can disrupt the cell redox balance towards a pro-oxidant state, leading to the changes in different metabolic activities and the morphological and physiological traits of plant species [52].

Pb concentrations in plant roots and shoots ranged from 123 to 678 and 29 to 287 mg kg$^{-1}$, respectively. Among the plant species, *B. lycium* Royle and *P. hysterophorus* L. accumulated the highest and lowest concentration of Pb, respectively (Table 2). Pb does not play any known metabolic and biological roles in plants, and plants are even equipped with an active defense mechanism against Pb stress that keeps its interaction in sensitive biological tissues. However, a mobile fraction of Pb present in soil can accumulate in plants and, thereby, enter the food chain. Total Pb concentration > 30 mg kg$^{-1}$ in plant tissues is considered toxic for plants species [53]. For most plant species, a higher level of Pb accumulation can reduce seed germination, the inhibition of chlorophyll synthesis, reduction in plant biomass, and negative impacts on enzymatic reactions and nutritional imbalance. One of

the major impacts of Pb toxicity on plants is the quick inhibition of the root cells' growth, which may be because of the blockage of cell division in root tips [54].

Zn concentrations varied in the plant roots and shoots at 28–456 and 12–321 mg kg$^{-1}$, respectively. *C. intybus* L. and *A. creticus* Lam. showed the highest and lowest Zn concentrations, respectively (Table 2). Although Zn is an essential micronutrient for plants, it becomes toxic at higher concentrations. Under normal conditions, Zn concentration weigh up to 60 mg kg$^{-1}$ dry weight; however, if this concentration increases and reaches 500 mg kg$^{-1}$ dry weight, it inhibits root elongation [55]. It has a long biological lifetime and is an important micronutrient that significantly affects metabolic activities of plants. The phytotoxicity of Zn reduces both plant root and shoot growth, leading to chlorosis in fresh younger leaves, and it can extend to mature leaves after prolonged exposure to higher Zn concentrations. At a higher level, Zn toxicity induces oxidative stress, which is usually very unstable and short-lived but chemically very reactive. The reactive oxygen species (ROS) produced inside plants as a result of Zn toxicity induces oxidative stress, which causes lipid peroxidation, membrane damage, and enzyme inactivation in the cell [56].

HM bioaccumulation in plants is controlled by multiple factors, such as plant species diversity, preferential uptake, and the binding of some metal species, growing stage, and elemental characteristics of soil [38]. Total phosphorous concentrations in the soil can also affects HMs uptake by plants. Moreover, the manner in which HMs interact with each other in soil defines the amount of particular metal that is taken up by the plant species. The enrichment of toxic HMs in indigenous plant species can lead to serious problems for the local community of the area, as it ultimately can end up in the food chain [57]. Hyperaccumulator plants must be protected, and the produced waste in the form of biomass should be treated in a separate chamber to restrict the HM's reach. It is high time to search for hypertolerance strategies to minimize and restrict HMs in the environment by retaining and detoxifying the underground parts of the plants.

*3.3. Phytoremediation Potential of the Studied Plants*

Plants possessing the ability to survive in metal-rich soils are known as metallophytes. Prolonged exposure of metallophytes to an excessive amount of HMs enables evolutions in their tolerances via a defensive mechanism and the development of a unique capacity to withstand, survive, and reproduce in a metal-rich environment [58]. Plants can be classified into three basic categories, excluder, accumulator, and hyperaccumulator, based on their responses when exposed to HMs.

Plants with BAF > 10, BAC > 1 and TF > 1 can be called hyperaccumulators [59]. The highest BAF values were observed for Pb (0.81), Zn (0.74), Fe (0.61), Cr (0.59), and Ni (0.54) in *A. maurorum* Medic., *A. creticus* Lam., *C. intybus* L., *P. hysterophorus* L., and *B. lycium* Royle, respectively (Table 3). Similarly, BAC values were the highest for Cr (11.19), followed by Zn (0.40), Ni (0.38), Pb (0.37), and Fe (0.23) in *A. creticus* Lam., *P. hysterophorus* L., *A. maurorum* Medic., and *P. hysterophorus* L., respectively. The maximum value of TF was for Ni (1.64) followed by Zn (0.62) in *P. hysterophorus* L. and *C. intybus* L., respectively, while the minimum value was for Cr (0.11) in *D. stramonium* L. (Table 3). TF and BAC values > 1 can be regarded as hyperaccumulators for the respective metals [44,60–62]. The results revealed that based on TF and BAC values, *A. maurorum* Medic., *A. creticus* Lam., *C. intybus* L., *B. lycium* Royle, and *D. stramonium* L. were hyperaccumulators for Cr while *P. hysterophorus* L. was promising species for both Ni and Cr. Our results demonstrate that *D. stramonium* L. was the most efficient species for Cr phytoextraction followed by *C. intybus* L. and *A. maurorum* Medic., respectively. From these results, it is very clear that the hyperaccumulation of different metals differs with different plant species tested, and these results are in line with other researchers [60–62].

**Table 3.** Phytoremediation potential factors of the studied plants.

| Plants | Cr | | | Ni | | | Fe | | | Pb | | | Zn | | |
|---|---|---|---|---|---|---|---|---|---|---|---|---|---|---|---|
| | BAF | BAC | TF | BAF | BAC | TF | BAF | BAC | TF | BAF | BAC | TF | BAF | BAC | TF |
| *A. creticus* Lam. | 0.38 | 6.74 | 0.39 | 0.15 | 0.08 | 0.54 | 0.36 | 0.17 | 0.48 | 0.47 | 0.09 | 0.18 | 0.74 | 0.40 | 0.54 |
| *A. maurorum* Medic. | 0.40 | 10.06 | 0.25 | 0.33 | 0.06 | 0.18 | 0.40 | 0.15 | 0.38 | 0.81 | 0.37 | 0.46 | 0.42 | 0.15 | 0.35 |
| *P. hysterophorus* L. | 0.59 | 8.61 | 0.20 | 0.45 | 0.38 | 1.64 | 0.60 | 0.23 | 0.39 | 0.53 | 0.25 | 0.47 | 0.44 | 0.16 | 0.36 |
| *B. lycium* Royle | 0.35 | 9.20 | 0.31 | 0.54 | 0.17 | 0.36 | 0.38 | 0.14 | 0.37 | 0.49 | 0.10 | 0.21 | 0.61 | 0.19 | 0.31 |
| *D. stramonium* L. | 0.49 | 11.19 | 0.18 | 0.41 | 0.11 | 0.26 | 0.28 | 0.10 | 0.36 | 0.48 | 0.05 | 0.11 | 0.54 | 0.11 | 0.20 |
| *C. intybus* L. | 0.40 | 11.06 | 0.23 | 0.48 | 0.26 | 0.53 | 0.61 | 0.14 | 0.23 | 0.52 | 0.18 | 0.34 | 0.45 | 0.28 | 0.62 |

*3.4. Pollution Indices*

Pollution indices such as the geoaccumulation index (Igeo), contamination factor (CF), and enrichment factor (EF) were used to gauge HMs pollution levels in the dumpsite soil of Peshawar (Table 4). The highest CF value was calculated in the case of Fe (41.86), followed by Ni (18.99), Pb (17.18), and Cr (13.16), while the lowest values were observed for Zn (4.22), designating very high contamination (CF > 6) for Fe, Ni, Pb, and Cr and considerable contamination (3 < CF < 6) from Zn. As shown in Figure 2, the degree of contamination values ranged from 35.08 to 78.76, showing a very high degree of contamination (DC > 24). Likewise, the highest geoaccumulation was observed for Fe (27.90), followed by Ni (12.66), Pb (11.45), and Cr (8.77), while the lowest value was observed for Zn (0.24), indicating that the soil was strong to extremely polluted (class 6, >5) for Fe, Ni, and Pb and low to moderately polluted (class 1, 0–1) for Zn (Table 4). Ni showed the highest EF value of 3.52, followed by Pb (3.38) and Cr (3.37), and the lowest value was demonstrated by Zn (2.04), depicting moderate enrichment (2 < EF ≤ 5) for all analyzed metals. The Eri values for the analyzed metals were Ni, (92.95), Pb (83.82), Fe (27.90), Cr (8.77), and Zn (2.86), showing moderate risks (40 < Eri ≤ 80) for Ni and Pb and low risks (Eri > 40) for Fe, Cr, and Zn (Table 4). RERI values ranged from 67.31 to 195.84, employing considerable risks from the dumpsite's soil (Figure 2). From these results, it has become clear that the dumpsite was contaminated with different heavy metals that have the potential to cause certain risks to human health through various mechanisms [63].

**Table 4.** Soil pollution indices of the dumpsite's soil.

| Sample | Contamination Factor | | | | | Geoaccumulation Index | | | | | Enrichment Factor | | | | Monomial Ecological Risk | | | |
|---|---|---|---|---|---|---|---|---|---|---|---|---|---|---|---|---|---|---|
| | Cr | Ni | Fe | Pb | Zn | Cr | Ni | Fe | Pb | Zn | Cr | Ni | Pb | Zn | Cr | Ni | Pb | Zn |
| S$_1$ | 3.52 | 4.25 | 41.86 | 7.34 | 0.35 | 2.33 | 3.17 | 27.9 | 4.89 | 0.24 | 0.48 | 1.65 | 0.68 | 0.08 | 7.76 | 23.27 | 36.69 | 0.35 |
| S$_2$ | 2.98 | 9.49 | 16.57 | 8.09 | 2.21 | 1.99 | 6.32 | 11.05 | 5.39 | 1.47 | 1.03 | 2.31 | 1.93 | 1.33 | 5.96 | 45.43 | 40.44 | 2.21 |
| S$_3$ | 5.72 | 5.30 | 16.43 | 5.29 | 3.35 | 3.83 | 3.21 | 10.95 | 3.53 | 2.24 | 1.98 | 0.02 | 1.25 | 2.04 | 11.4 | 28.51 | 26.47 | 3.35 |
| S$_4$ | 13.16 | 14.79 | 22.29 | 16.76 | 4.22 | 8.77 | 9.52 | 14.86 | 11.18 | 2.86 | 3.37 | 0.04 | 2.92 | 1.88 | 26.31 | 74.96 | 83.82 | 4.25 |
| S$_5$ | 6.92 | 18.99 | 20.20 | 17.18 | 3.17 | 4.61 | 12.66 | 13.47 | 11.45 | 2.11 | 1.96 | 3.52 | 3.38 | 1.57 | 13.84 | 92.95 | 85.88 | 3.17 |
| S$_6$ | 10.89 | 14.79 | 21.54 | 11.51 | 3.77 | 7.26 | 9.53 | 14.36 | 7.68 | 2.51 | 2.54 | 1.56 | 3.76 | 1.43 | 21.78 | 74.96 | 57.57 | 3.77 |

Such higher values for HMs consolidate our hypothesis that the dumpsite is illegally in use for both hazardous and non-hazardous waste, which is not technically and legally allowed otherwise. Wastes such as biosolids and manures, e.g., livestock manures, compost, and municipal sewage sludge, when disposed of in open dumpsites, can cause HMs accumulation such as As, Cd, Cr, Pb, Hg, Se, Zn, Ti, Sb, and so forth in the soil. Although most of organic waste contains a lower amount of HMs, continuous dumping can lead to HM accumulation in soil. In the current study, the higher pollution indices values for Ni, Cr, Pb, and Fe can be regarded as industrial waste, incinerated waste from hospitals, barbershop wastes, and other mixed types of waste streams coming from the local community.

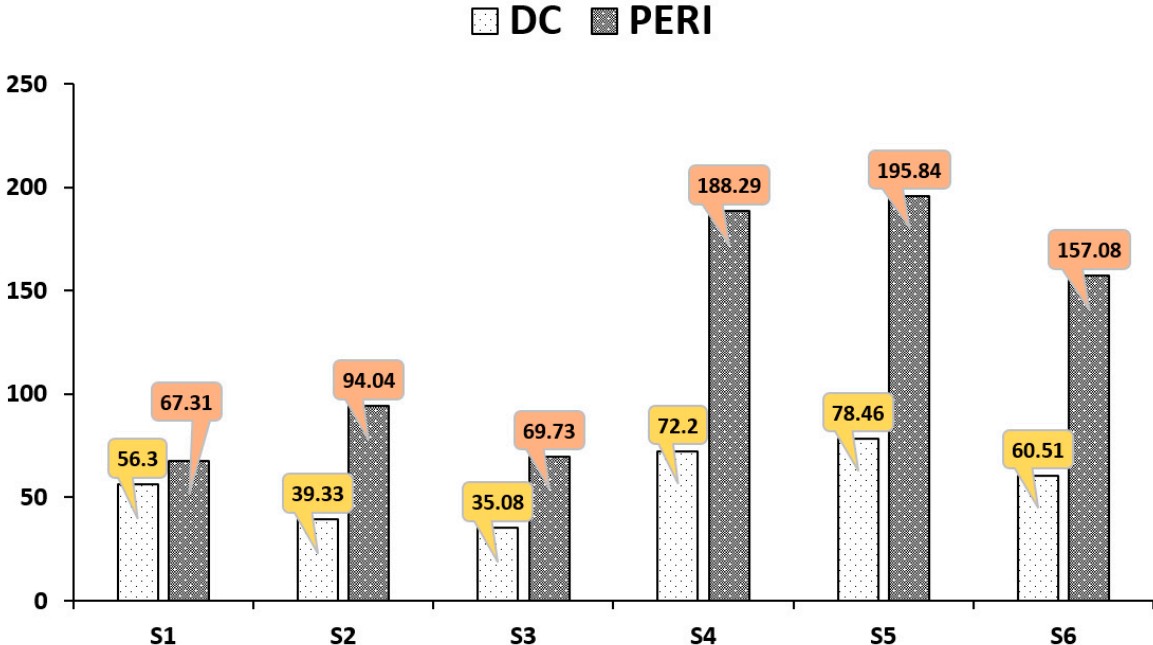

**Figure 2.** Degree of contamination (DC) and potential ecological risk index (PERI) of the collected samples.

*3.5. Human Health Risk Assessment*

3.5.1. Average Daily Dose (ADDs) from Soil Exposure

Many researchers have brought up the connection between open dumpsite pollution and health problems. People who live or work near open dumps have greater chances of congenital birth defects, cancer, and respiratory illness. The evidence linking waste in open dumps and incinerators to health problems such as cancer, and fertility conditions has long been reported [64,65]. To assess the human health hazard from dump site soil exposure, the non-cancerous effects of Cr, Ni, Pb, Fe, and Zn and the carcinogenic risks of Cr, Pb, and Ni via three major pathways called inhalation, ingestion, and dermal interaction were investigated (Tables 5 and 6).

The ingestion route poses the highest risks, followed by dermal interactions and the inhalation of the soil's particles. The same trend has been reported by previous researchers [31–33,65]. The highest and lowest values in $ADD_{ing}$ were for Pb ($1.38229 \times 10^{-6}$ and Cr ($9.4333 \times 10^{-5}$) in children and adults, respectively. Similarly, the maximum and minimum values of $ADD_{inh}$ were for Cr ($1.08392 \times 10^{-8}$) and Zn ($9.48432 \times 10^{-16}$) for children and adults, respectively (Table 5). Likewise, the highest and lowest values of $ADD_{derm}$ were for Pb ($1.49236 \times 10^{-10}$) and Ni ($8.57708 \times 10^{-10}$) in children, respectively. In the current study, the $ADD_{ing}$, $ADD_{inh}$, and $ADD_{derm}$ values were found to be lower than the respective RfD values. HMs were in a decreasing order in $ADD_{ing}$ (Pb > Ni > Fe > Zn > Cr), $ADD_{inh}$ (Cr > Ni > Pb > Fe > Zn), and $ADD_{derm}$ (Pb > Fe > Zn > Cr > Ni) in both adults and children(Table 5). The maximum and minimum LADD values were for Fe ($9.83827 \times 10^{-16}$) and Cr ($1.7844 \times 10^{-8}$) in both children and adults, respectively. The assessed HMs were in decreasing order in LADD (Cr > Zn > Fe > Ni > Pb) (Table 5).

**Table 5.** Average daily dose (ADD) and estimated lifetime average daily dose (LADD) of soil results for adults and children in the study area.

| PTEs | ADD$_{ing}$ | | | ADD$_{inh}$ | | | ADD$_{derm}$ | | | LADD | |
|------|-------|----------|-----|-------|----------|-----|-------|----------|-----|-------|----------|
| | Adult | Children | RfD | Adult | Children | RfD | Adult | Children | RfD | Adult | Children |
| Cr | $9.4333 \times 10^{-5}$ | $6.47144 \times 10^{-6}$ | $3.00 \times 10^{-3}$ | $1.03736 \times 10^{-16}$ | $1.08392 \times 10^{-8}$ | $2.86 \times 10^{-5}$ | $2.3937 \times 10^{-7}$ | $6.98675 \times 10^{-10}$ | $5.00 \times 10^{-5}$ | $4.04615 \times 10^{-15}$ | $1.7844 \times 10^{-8}$ |
| Ni | $8.0001 \times 10^{-3}$ | $7.94447 \times 10^{-6}$ | $2.00 \times 10^{-2}$ | $1.27349 \times 10^{-16}$ | $1.33065 \times 10^{-8}$ | $2.20 \times 10^{-4}$ | $2.93856 \times 10^{-7}$ | $8.57708 \times 10^{-10}$ | $1.20 \times 10^{-2}$ | $4.96713 \times 10^{-15}$ | $2.19057 \times 10^{-8}$ |
| Fe | $8.2433 \times 10^{-5}$ | $5.65504 \times 10^{-6}$ | $8.40 \times 10^{2}$ | $9.06495 \times 10^{-17}$ | $9.47182 \times 10^{-9}$ | $2.20 \times 10^{-4}$ | $2.09173 \times 10^{-7}$ | $6.10535 \times 10^{-10}$ | $6.00 \times 10^{-2}$ | $9.83827 \times 10^{-16}$ | $4.07993 \times 10^{-9}$ |
| Pb | $2.0149 \times 10^{-5}$ | $1.38229 \times 10^{-6}$ | $3.50 \times 10^{-3}$ | $2.21578 \times 10^{-17}$ | $2.31523 \times 10^{-9}$ | $3.52 \times 10^{-2}$ | $5.11289 \times 10^{-8}$ | $1.49236 \times 10^{-10}$ | $5.25 \times 10^{-4}$ | $8.64249 \times 10^{-16}$ | $3.81145 \times 10^{-9}$ |
| Zn | $8.6246 \times 10^{-5}$ | $8.91666 \times 10^{-6}$ | $3.00 \times 10^{-1}$ | $9.48432 \times 10^{-17}$ | $9.91001 \times 10^{-9}$ | $2.06 \times 10^{-2}$ | $2.1885 \times 10^{-7}$ | $6.3878 \times 10^{-10}$ | $7.00 \times 10^{-2}$ | $8.68004 \times 10^{-16}$ | $1.49957 \times 10^{-8}$ |

PTEs = potentially toxic elements; ADD$_{ing}$ = average daily dose through ingestion; ADD$_{inh}$ = average daily dose through inhalation; ADD$_{derm}$ = average daily dose through the skin; LADD = estimated lifetime average daily dose.

**Table 6.** Non-carcinogenic and carcinogenic exposure from dumpsite soil in the study area.

| PTEs | Non-Carcinogenic Risk | | | | | | Carcinogenic Risk | |
|------|-------|----------|-------|----------|-------|----------|-------|----------|
| | HRI$_{ing}$ | | HRI$_{inh}$ | | HRI$_{derm}$ | | CRI | |
| | Adult | Children | Adult | Children | Adult | Children | Adult$_{ing}$ | Children$_{ing}$ |
| Cr | $4.90623 \times 10^{-6}$ | $4.49 \times 10^{-7}$ | $5.39528 \times 10^{-18}$ | $4.50995 \times 10^{-10}$ | $6.22477 \times 10^{-9}$ | $6.18708 \times 10^{-10}$ | $3.43923 \times 10^{-17}$ | $1.9989 \times 10^{-8}$ |
| Ni | $5.79026 \times 10^{-6}$ | $5.3 \times 10^{-7}$ | $6.36744 \times 10^{-18}$ | $5.32258 \times 10^{-10}$ | $7.34639 \times 10^{-9}$ | $7.3019 \times 10^{-10}$ | $4.22206 \times 10^{-17}$ | $1.84008 \times 10^{-8}$ |
| Fe | $4.12163 \times 10^{-6}$ | $3.77 \times 10^{-7}$ | $4.53248 \times 10^{-18}$ | $3.78873 \times 10^{-10}$ | $5.22932 \times 10^{-9}$ | $5.19765 \times 10^{-10}$ | N/A | N/A |
| Pb | $1.00747 \times 10^{-6}$ | $9.22 \times 10^{-8}$ | $1.10789 \times 10^{-18}$ | $9.26094 \times 10^{-11}$ | $1.27822 \times 10^{-9}$ | $1.27048 \times 10^{-10}$ | $7.34611 \times 10^{-18}$ | $3.20162 \times 10^{-9}$ |
| Zn | $4.31231 \times 10^{-6}$ | $3.94 \times 10^{-7}$ | $4.74216 \times 10^{-18}$ | $3.964 \times 10^{-10}$ | $5.47124 \times 10^{-9}$ | $5.43811 \times 10^{-10}$ | N/A | N/A |
| **HI** | $2.01379 \times 10^{-5}$ | $1.84 \times 10^{-6}$ | $2.21452 \times 10^{-17}$ | $1.85114 \times 10^{-9}$ | $2.55499 \times 10^{-8}$ | $2.53952 \times 10^{-9}$ | | |

PTEs = potentially toxic elements; HRI$_{ing}$ = health risk index through ingestion; HRI$_{inh}$ = health risk index through inhalation; HRI$_{derm}$ = health risk index through skin; HI = health index; CRI = carcinogenic risk index.

### 3.5.2. Carcinogenic and Non-Carcinogenic Risks

Table 6 summarizes carcinogenic and non-carcinogenic health risks and health hazards from HMs in the soil in both children and adults. For non-carcinogenic risks, the maximum value was for Pb ($1.00747 \times 10^{-6}$) and Fe ($3.77 \times 10^{-7}$), and the minimum value was for Ni ($5.79026 \times 10^{-6}$) and Pb ($9.22 \times 10^{-8}$) in adults and children, respectively. The values were in an decreasing order for adults (Pb > Fe > Zn > Cr > Ni) and for children (Fe > Zn > Cr > Ni > Pb), respectively. In the same manner, the maximum value for non-carcinogenic $HRI_{inh}$ was for Pb ($1.10789 \times 10^{-18}$) and Fe ($3.78873 \times 10^{-10}$), while the minimum values were for Ni ($6.36744 \times 10^{-18}$) and Pb ($9.26094 \times 10^{-11}$) for both adults and children, respectively. A decreasing trend in non-carcinogenic $HRI_{ing}$ values was observed for adults (Pb > Fe > Zn > Cr > Ni) and for children (Fe > Zn > Cr > Ni > Pb). Likewise, $HRI_{derm's}$ highest values were for Pb ($1.27822 \times 10^{-9}$) and (1.27048E-10), and the minimum values were Ni ($7.34639 \times 10^{-9}$, $7.3019 \times 10^{-10}$) in both adults and children, respectively. The $HRI_{derm}$ value's decreasing trend was (Pb > Fe > Zn > Cr > Ni) for both adults and children. The HRI values > 1 are considered significant for non-carcinogenic risks. As the followed criteria for non-carcinogenic risks, none of the values for the analyzed metals exceeded the threshold's limits. However, municipality workers and other local waste pickers are vulnerable to non-carcinogenic risks because of the frequent contact with soil.

The carcinogenic health risk was calculated for both adults and children regarding Cr, Pb, and Ni via the ingestion route only in the study area. The maximum value was in the case of Cr ($3.43923 \times 10^{-17}$), followed by Ni ($1.84008 \times 10^{-8}$), while the minimum value was in Pb ($7.34611 \times 10^{-18}$), ($3.20162 \times 10^{-9}$) for adults and children, respectively (Table 6). The CRI values between $1 \times 10^{-6}$ and $1 \times 10^{-4}$ are acceptable, while values $> 1 \times 10^{-4}$ are considered significant for cancer risks. The current study's results demonstrate that none of the CRA values exceeded the safe limit. The CRA values for Cr and Ni were relatively higher in children and, hence, can pose risks in case of prolonged exposure. Excessive concentrations of Cr can be extremely toxic and may cause neurodegenerative changes, including diseases such as Alzheimer's. Similar toxic effects of Ni on the reproductive and immune systems were reported if taken beyond the normal limits [66].

### 3.5.3. Statistical Analysis

Pearson's correlation, principle component analysis (PCA), and ANOVA were applied for data analysis. Correlation analysis was applied to analyze the association between HMs in soil and different parts of the plant species (Figure 3). The cross sign in Figure 3 shows the relationship of the parameters as non-significant at ($p < 0.05$), while blue and red colors show a positive and negative correlation, respectively. White squares represent the main diagonal line separating the upper and lower triangles. Soil Cr, Zn, and Fe showed a non-significant correlation ($p < 0.05$) with each other, suggesting different sources in the dumpsite's soil, while soil Ni and Pb had the highest correlation, showing the same source of disposal. Soil Ni and Pb showed a non-significant correlation with Cr, Zn, and Fe.

PCA was conducted for source identification and the association of HMs in different parts of the studied plants. PCA is regarded as an effective and accurate tool for source identification, dimensionality reduction, and data visualization [67]. The data were run through Varimax rotation with Kaiser Normalization to measure mean sampling adequacy, which turned a complex matrix generated from a combination of measured parameters into a significant pattern, and the obtained loading factors and Biplot were used for data interpretation. Before submitting the data for PCA analysis, Kaiser–Meyer–Olkin (KMO) and Bartlett's sphericity index tests were performed using SPSS. The KMO and Bartlett's sphericity values were 0.61 and $3.3 \times 10^{-18}$, which suggested that there was a less partial correlation compared to the sum of correlations; hence, the data were subjected to PCA. KMO values between 0.8 and 1 show that the sampling is adequate, less than 0.6 indicates that sampling is inadequate, and values closer to zero mean large partial correlations in the

data or widespread correlation, which is a limiting factor for PCA. Initially, PCA resolves into the same number of components as the given variables, which could be reduced based on the Eigenvalue (i.e., cut off eigenvalue > 1) for the selected components.

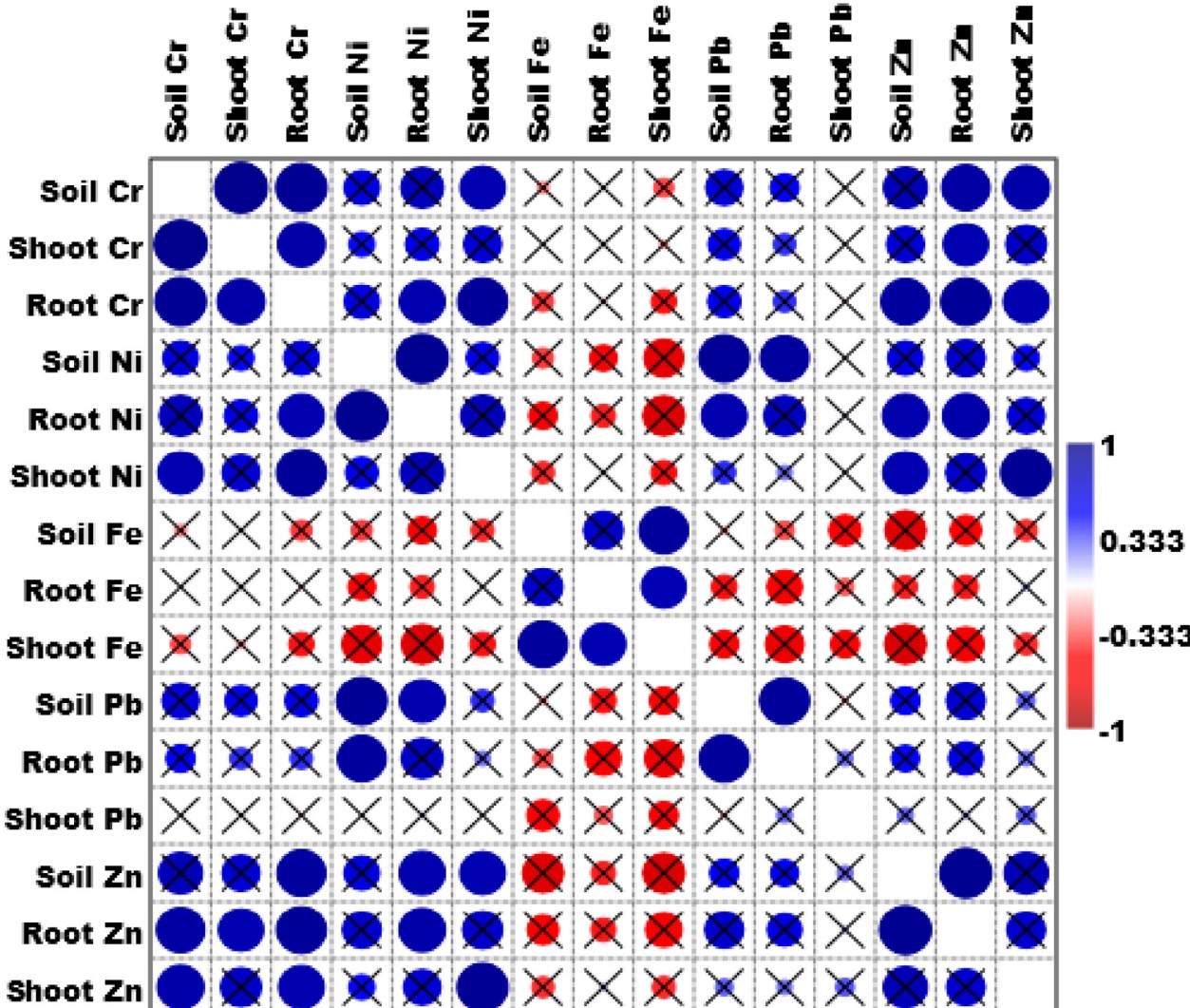

**Figure 3.** Pearson's correlation showing the relationship among heavy metals in different parts of plants and soil. Where "X" shows that there no relation between the tested variables.

Based on the adopted criteria, five PCs with eigenvalues > 1 were selected from PCA analysis, contributing an accumulative variance of 83.16% (Table 7). $PC_1$ contributed 45.14% variation and was dominated by Ni concentration in root and shoots of the studied plants, and a weak correlation was found between Cr and Ni concentration in soil. A strong correlation between root and shoots indicates a higher mobilization of the metal from the substrate part to aerial part, suggesting good potential for phytoextraction.

Factors such as pH, temperature, moisture, aeration, competition between species, elemental bioavailability, leave types, root system, plant size, and type of plants strongly influence metal absorption and uptake in plants [68]. Ni has low phytoavailability and mobility in normal soil. Soil factors such as pH and OM significantly influence Ni accumulation in the soil–plant system. Due to the slightly acidic to alkaline soil, higher OM contents, and EC in the dumpsite, Ni concentrations were higher in roots and shoots. Similarly, in $PC_2$ (12.90%), a strong correlation was found between Cr concentrations in roots to shoots, and a weak correlation was observed in soil Fe and Cr. Moreover, in $PC_2$, soil EC was weakly correlated with both soil Fe and Cr in plant roots and shoots. The reason for a strong

correlation between Cr concentrations in plant roots to shoots ratio is because absorption mainly depends on the speciation of the metal, which determines its uptake, translocation, and accumulation; however, its detailed mechanism is not fully understood. Cr entry in root cells may occur through the entry channels of the essential ions [69]. Cr translocation mechanisms from root to shoots need to be investigated in detail at the cellular level. In $PC_3$ (9.14%), $PC_4$ (8.46%), and $PC_5$ (7.50%), soil pH and EC were positively correlated with soil metals (Pb, Cr, Fe, Zn, and Ni), consolidating the previous literature (Table 7).

**Table 7.** Factor loadings for studied physiochemical parameters of the dumpsite's soil and heavy metal distribution in plants.

| Variables | PC 1 | PC 2 | PC 3 | PC 4 | PC 5 |
|---|---|---|---|---|---|
| pH | −0.08 | −0.05 | **0.23** | 0.04 | **0.30** |
| EC ($\mu$S cm$^{-1}$) | −0.25 | **0.16** | −0.32 | **0.26** | **0.38** |
| TDS mg/L | −0.31 | −0.10 | −0.09 | 0.06 | 0.03 |
| Bulk density g/cm$^3$ | −0.23 | 0.19 | −0.09 | **0.33** | 0.03 |
| Soil Cr | **0.12** | **0.21** | −0.01 | −0.07 | −0.11 |
| Soil Ni | **0.06** | −0.14 | −0.02 | 0.02 | **0.17** |
| Soil Fe | −0.26 | **0.18** | 0.013 | 0.06 | −0.03 |
| Soil Pb | 0.04 | 0.07 | −0.214 | **0.29** | −0.62 |
| Soil Zn | **0.30** | −0.08 | **0.40** | **0.21** | **0.26** |
| Root Cr | 0.04 | **0.29** | −0.05 | 0.06 | 0.18 |
| Shoot Cr | 0.01 | **0.26** | 0.012 | −0.43 | 0.05 |
| Root Ni | **0.38** | −0.06 | −0.30 | −0.15 | −0.03 |
| Shoot Ni | **0.24** | 0.08 | −0.33 | −0.24 | 0.09 |
| Root Fe | −0.24 | 0.10 | 0.11 | −0.46 | −0.05 |
| Shoot Fe | −0.38 | −0.06 | 0.34 | −0.12 | −0.19 |
| Root Pb | −0.07 | −0.44 | −0.29 | 0.05 | −0.18 |
| Shoot Pb | −0.01 | −0.63 | −0.01 | −0.18 | 0.07 |
| Root Zn | 0.19 | −0.07 | 0.27 | 0.31 | −0.03 |
| Shoot Zn | 0.12 | 0.12 | 0.33 | −0.02 | −0.34 |
| Eigenvalue | 3.53 | 1.79 | 1.41 | 1.28 | 1.05 |
| Variance | 45.14 | 12.90 | 9.14 | 8.46 | 7.50 |
| **Accumulative variance** | **45.14** | **58.04** | **67.19** | **74.65** | **83.16** |

PCA Biplot was used to visualize the plants' potential in terms of HM uptake. A biplot was mostly used to display the data matrix graphically. In Figure 4, the convex hull shows the overall metal concentration taken up by the analyzed plant species in both roots and shoots. The convex hull elongation indicates the standard deviation from the mean while the distance from the center shows the concentration of the metals along the associated PC. In Figure 4, $PC_1$ contributed 48.50% of total variation, followed by $PC_2$ with 19.30% variation, accumulatively contributing 67.80%. The standard deviation (SD) was higher in *C. intybus* L. and *P. hysterophorus* L., while a lower SD was observed in *B. lycium Royle*. Furthermore, *A. creticus* Lam. showed a stronger association with $PC_1$, indicating higher influences or the highest overall metal uptake while *A. maurorum* Medic. showed lower associations with $PC_1$, representing a low influence or fewer amounts of the overall metal uptake. *A. creticus* Lam. and *A. maurorum* Medic. showed independent behaviors, while *C. intybus* L, *B. lycium* Royle, *D. stramonium* L., and *B. lycium* Royle were similar in some aspects. The convex hull overlapping in the case of *C. intybus* L., *P. hysterophorus* L., *D. stramonium* L., and *B. lycium* Royle shows that there is no significant difference among the species in terms of overall HM uptake.

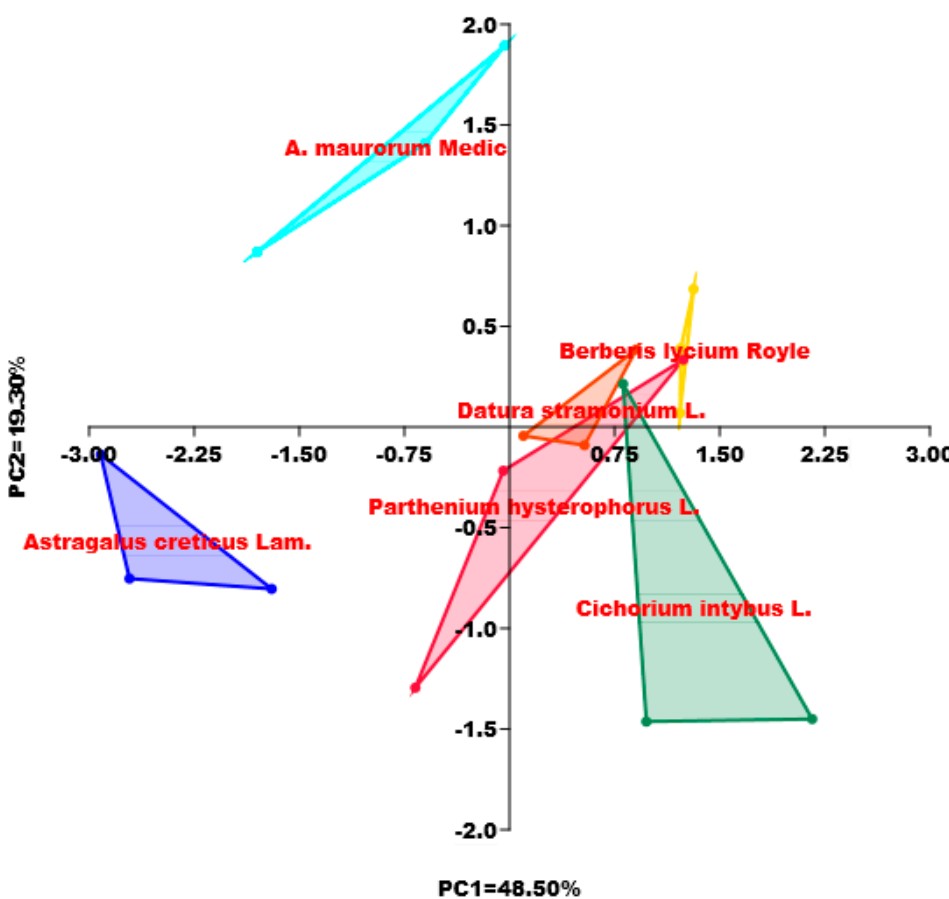

**Figure 4.** Biplot diagram of PCA showing plant behavior and potentials of overall metals accumulation.

## 4. Conclusions

The current study investigated the phytoremediation potential of native plants, heavy metal (HM) contamination statuses in plants and soil, and their associated health risks from a major municipal open dump site in Peshawar city. The results demonstrated that based on TF and BAC values, *A. maurorum* Medic., *A. creticus* Lam., *C. intybus* L, *B. lycium* Royle, and *D. stramonium* L. were hyperaccumulators for Cr, while *P. hysterophorus* L. was promising species for both Ni and Cr. The municipal solid waste's composition dictates the HM input and distribution in both plants and soil. Among various studied parameters, CF, Igeo, EF, and Eri values were found higher for Fe, Ni, Pb, and Cr, showing high contamination, extreme pollution, moderate enrichment, and moderate risk while Zn showed considerable contamination, moderate pollution, moderate enrichment, and low risk in dumpsite soils. PERI values showed that the dumpsite's soil can cause considerable risks. Moreover, both adults and children were highly exposed to Pb via $ADD_{ing}$ and $ADD_{derm}$, while the rate of exposure was higher in Cr via $ADD_{inh}$. The exposure level was found lower for Fe, Ni, and Zn, respectively. LADD values were higher for Fe and Cr in both adults and children showing a greater risk for carcinogenic diseases. Non-carcinogenic exposure was higher from Pb and Cr and lower from Ni and Fe in both adults and children, respectively. In addition, Cr and Ni showed the highest carcinogenic exposure, while Pb showed the lowest carcinogenic exposure for both adults and children, respectively. Finally, it is expected that the current study's findings would serve as a database for future monitoring of the municipal waste dumpsite in Pakistan. It will also help to support the implementation of public policies to ensure sustainability and developmental activities in the study's area. Therefore, to minimize the potential health problems arising from the dumpsite, it is substantive that special attention should be paid to minimizing and overcoming the health impacts and work for sustainable and eco-friendly remedial measures. In future, it is

possible to assess the effectiveness of various management measures and assure the safety of the world's food supply by developing an effective and straightforward metal transport model for prediction.

**Author Contributions:** Conceptualization, M.S., A.D. and S.U.; data curation, M.S., A.D., H.U., A.K. and S.U.; formal analysis, E.B.-G., A.D., H.U., A.K. and S.U.; funding acquisition, T.K.F.; investigation, M.S. and A.K.; methodology, E.B.-G., A.D. and A.K.; project administration, E.B.-G. and A.D.; resources, E.B.-G., A.D. and A.K.; software, A.D., H.U., A.K. and S.U.; supervision, A.D.; validation, A.D., H.U., A.K. and S.U.; visualization, M.S., E.B.-G., A.D., H.U., T.K.F., A.K. and S.U.; writing—original draft, M.S. and H.U.; writing—review and editing, M.S., E.B.-G., A.D., H.U., T.K.F., A.K. and S.U. All authors have read and agreed to the published version of the manuscript.

**Funding:** This work funded by Researchers Supporting Project (no. RSP2022R487), King Saud University, Riyadh, Saudi Arabia.

**Institutional Review Board Statement:** Not applicable.

**Informed Consent Statement:** Not applicable.

**Data Availability Statement:** Not applicable.

**Acknowledgments:** We extend our appreciation to the Researchers Supporting Project at King Saud University, Riyadh, Saudi Arabia, for funding this research project, (Fund no. RSP2022R487).

**Conflicts of Interest:** The authors declare no conflict of interest.

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
