# Peer review of "Bioaccumulation of Heavy Metals in a Soil–Plant System from an Open Dumpsite and the Associated Health Risks through Multiple Routes"

_sustainability, doi:10.3390/su142013223_

Round 1
Reviewer 1 Report
In this manuscript, the authors studied the contamination due to heavy metals in a specific area. Moreover, they tested different plants for their phytoremediation potential of five different heavy metals. Finally, they studied a high amount of different pollution indices, including some of them related with human health risks and found different correlations between heavy metals concentrations in the dumpsite area. The results are promising due to the accumulation capacity of the different plants. However, some results are not properly discussed and these parts of the manuscript seems to be enhanced in much clearer discussion.
Herein, it is suggested that the manuscript could be reconsider for publication in Sustainability after minor corrections has been conducted according to the recommended points.
Required corrections:
Introduction:
i) Line 94-98: The first sentence includes “various studies” and there is not any reference. However, the second sentence includes that “few or no studies” and there are 3 references. It would be clear if the references are at the end of the first sentence or if the authors rephrase this.
ii) Line 102-105: The purpose of the study should be better explained, specially why they selected these five heavy metals.
Materials and Methods:
iii) Line: 136: Change HCLO4 for HClO4.
iv) Line 155-159: Include the units of the concentration values in the different equations or in the text. Similar with the other equations where the units are not specified in the text.
Results and Discussion:
v) Line 289: It is suggested that the discussion of Table 1 results may be included after explained the results of Table 1. This would make the document easier to read than now, that the discussion of Table 1 is after the results and discussion of Table 2 (Line 318-334)
vi) Line 335: In 3.2 there is a description of the results and a discussion about the effect of these heavy metals in plants metabolism. However, there is not any comparison with previous accumulation studies in plants. It would be good to know if the accumulation capacity of these plants in roots and shoots is similar, higher or lower than in other plants previously studied.
vii) Line 402: The usual order of heavy metals is Cr, Ni, Fe, Pb and Zn. However, in Table 3 it is Cr, Fe, Zn, Ni and Pb. Please reorder them in order to be consistent in all the manuscript.
viii) Line 420: In 3.3 the results are well described and discussed between them. However, as Sustainability instruction of authors says: “authors should discuss the results and how they can be interpreted in perspective of previous studies and of the working hypotheses”. Please, include the discussion.
ix) Line 441: In 3.4 the results are well described and discussed between them. However, as Sustainability instruction of authors says: “authors should discuss the results and how they can be interpreted in perspective of previous studies and of the working hypotheses”. Please, include the discussion.
x) Line 480: There are two 60 numbers in the reference. It is suggested to checl where is the problem in this reference.
Author Response
Response to the Reviewer 1 comments
In this manuscript, the authors studied the contamination due to heavy metals in a specific area. Moreover, they tested different plants for their phytoremediation potential of five different heavy metals. Finally, they studied a high amount of different pollution indices, including some of them related with human health risks and found different correlations between heavy metals concentrations in the dumpsite area. The results are promising due to the accumulation capacity of the different plants. However, some results are not properly discussed and these parts of the manuscript seems to be enhanced in much clearer discussion. Herein, it is suggested that the manuscript could be reconsider for publication in Sustainability after minor corrections has been conducted according to the recommended points.
Response: Thanks for your recommendation. We have improved the mentioned portion as per your comments
Required corrections:
Introduction:
- i) Line 94-98: The first sentence includes “various studies” and there is not any reference. However, the second sentence includes that “few or no studies” and there are 3 references. It would be clear if the references are at the end of the first sentence or if the authors rephrase this.
Response: We apologize for this typo. The references were added to the next sentence (Please see line 105)
- ii) Line 102-105: The purpose of the study should be better explained, specially why they selected these five heavy metals.
Response: We have modified the mentioned portion and added discussion regarding the selection of the studied metals (Please see lines 111-115)
Materials and Methods:
iii) Line: 136: Change HCLO4 for HClO4.
Response: Changed (Please see line 145)
- iv) Line 155-159: Include the units of the concentration values in the different equations or in the text. Similar with the other equations where the units are not specified in the text.
Response: We have modified the mentioned equations by adding the units and highlighted them as yellow where modification was done
Results and Discussion:
- v) Line 289: It is suggested that the discussion of Table 1 results may be included after explained the results of Table 1. This would make the document easier to read than now, that the discussion of Table 1 is after the results and discussion of Table 2 (Line 318-334)
Response: We have placed Table 2 as per the suggestion of the reviewer
- vi) Line 335: In 3.2 there is a description of the results and a discussion about the effect of these heavy metals in plants metabolism. However, there is not any comparison with previous accumulation studies in plants. It would be good to know if the accumulation capacity of these plants in roots and shoots is similar, higher or lower than in other plants previously studied.
Response: We have modified the discussion portion as per the suggestion of the reviewer (Please see lines 353-356)
vii) Line 402: The usual order of heavy metals is Cr, Ni, Fe, Pb and Zn. However, in Table 3 it is Cr, Fe, Zn, Ni and Pb. Please reorder them in order to be consistent in all the manuscript.
Response: We have modified Table 3 as per the suggestion of the reviewer
viii) Line 420: In 3.3 the results are well described and discussed between them. However, as Sustainability instruction of authors says: “authors should discuss the results and how they can be interpreted in perspective of previous studies and of the working hypotheses”. Please, include the discussion.
Response: We have modified the discussion portion as per the suggestion of the reviewer (Please see lines 443-444 and 449-451)
- ix) Line 441: In 3.4 the results are well described and discussed between them. However, as Sustainability instruction of authors says: “authors should discuss the results and how they can be interpreted in perspective of previous studies and of the working hypotheses”. Please, include the discussion.
Response: We have modified the discussion portion as per the suggestion of the reviewer (Please see lines 467-469)
- x) Line 480: There are two 60 numbers in the reference. It is suggested to check where the problem in this reference is.
Response: Thanks for your comment. We have revisited all the references and corrected

Reviewer 2 Report
Manuscript Number: Sustainability-1903569
Decision: Minor Revisions
Comments
Abstract:
· Need and purpose of this study is missing in abstract.
· Novelty of the study is missing in the abstract.
· The abstract should briefly state the purpose of the research, the principal results and major conclusions. An abstract is often presented separately from the article, so it must be able to stand alone. This section isn't clear. Authors just collecting some ideas. Please, try to improve this section by highlighting the research gap and the novelty of this work. Also, try to lead the reader smoothly to your point.
Introduction
· Introduction structure is really smooth and leads to the problem well.
· Please just revise the introduction grammatically.
Methods
· Why Peshawar was selected as a study area, as Karachi is considered as the most populated city and come with the most open dumpsite garbage. Or may be you could have selected Sadiqabad, where there are many fertilizer industries which could have give you better study calculations ?
Results and Discussion
· It is suggested to compare the results of the present research with some similar studies which is done before.
· The Results and Discussion section is devoted, in large, by representing the research out comes' yielded, but a critical and integrated approach of these outcomes has been made, probable at a distinct "synthesis' and cross-cited subsection. In this distinct subsection the key-aspects that determine the outcomes have to be signified into a descriptive manner.
· Authors should support their conclusion from result with references and also compare the results with previous literature.
Conclusion
· Please make sure your conclusions' section underscore the scientific value added of your paper, and/or the applicability of your findings/results, as indicated previously. Please revise your conclusion part into more details. Basically, you should enhance your contributions, limitations, underscore the scientific value added of your paper, and/or the applicability of your findings/results and future study in this session.
Author Response
Response to the Reviewer 2 comments
Manuscript Number: Sustainability-1903569
Decision: Minor Revisions
Comments
Abstract:
Need and purpose of this study is missing in abstract. Novelty of the study is missing in the abstract. The abstract should briefly state the purpose of the research, the principal results and major conclusions. An abstract is often presented separately from the article, so it must be able to stand alone. This section isn't clear. Authors just collecting some ideas. Please, try to improve this section by highlighting the research gap and the novelty of this work. Also, try to lead the reader smoothly to your point.
Response: Thanks for your comments. We have added the need and purpose of the study and highlighted the novelty in the present study (Please see lines 29-33)
Introduction
Introduction structure is really smooth and leads to the problem well. Please just revise the introduction grammatically.
Response: Thanks for your appreciation. We have thoroughly checked and corrected any grammatical mistakes and highlighted them as yellow
Methods
Why Peshawar was selected as a study area, as Karachi is considered as the most populated city and come with the most open dumpsite garbage. Or may be you could have selected Sadiqabad, where there are many fertilizer industries which could have give you better study calculations ?
Response: As discussed in the research area section, Peshawar is also populous so municipal wastes, agriculture waste, household garbage, food waste, and other industrial waste are being disposed of without proper physical and chemical treatment in the open dump sites. Also, our focus was to explore the potential of hyperaccumulators from a dumpsite which was easily available in Peshawar
Results and Discussion
It is suggested to compare the results of the present research with some similar studies which is done before. The Results and Discussion section is devoted, in large, by representing the research out comes' yielded, but a critical and integrated approach of these outcomes has been made, probable at a distinct "synthesis' and cross-cited subsection. In this distinct subsection the key-aspects that determine the outcomes have to be signified into a descriptive manner. Authors should support their conclusion from result with references and also compare the results with previous literature.
Response: We have modified this section per the suggestions of both reviewers (Please see the text highlighted as yellow in this section)
Conclusion
Please make sure your conclusions' section underscore the scientific value added of your paper, and/or the applicability of your findings/results, as indicated previously. Please revise your conclusion part into more details. Basically, you should enhance your contributions, limitations, underscore the scientific value added of your paper, and/or the applicability of your findings/results and future study in this session.
Response: We have modified the conclusion section as per the suggestion of the reviewer (Please see lines 642-650)
